# Btk Inhibitors: A Medicinal Chemistry and Drug Delivery Perspective

**DOI:** 10.3390/ijms22147641

**Published:** 2021-07-16

**Authors:** Chiara Brullo, Carla Villa, Bruno Tasso, Eleonora Russo, Andrea Spallarossa

**Affiliations:** Department of Pharmacy, University of Genova, Viale Benedetto XV, 3, 16132 Genova, Italy; brullo@difar.unige.it (C.B.); villa@difar.unige.it (C.V.); tasso@difar.unige.it (B.T.)

**Keywords:** Bruton’s kinase, Btk inhibitors, ibrutinib, drug delivery, cancer treatment

## Abstract

In the past few years, Bruton’s tyrosine Kinase (Btk) has emerged as new target in medicinal chemistry. Since approval of ibrutinib in 2013 for treatment of different hematological cancers (as leukemias and lymphomas), two other irreversible Btk inhibitors have been launched on the market. In the attempt to overcome irreversible Btk inhibitor limitations, reversible compounds have been developed and are currently under evaluation. In recent years, many Btk inhibitors have been patented and reported in the literature. In this review, we summarized the (ir)reversible Btk inhibitors recently developed and studied clinical trials and preclinical investigations for malignancies, chronic inflammation conditions and SARS-CoV-2 infection, covering advances in the field of medicinal chemistry. Furthermore, the nanoformulations studied to increase ibrutinib bioavailability are reported.

## 1. Introduction

Bruton’s tyrosine kinase (Btk), also known as agammaglobulinemia tyrosine kinase (TK), is a member of the Tec kinase family, initially identified as a defective protein in human X-linked agammaglobulinemia (XLA) in 1993 by Vetrie and coworkers [1,2]. Btk is a cytoplasmic non-receptor TK expressed in all cells of the hematopoietic lineage, particularly B cells, mast cells and macrophages; on the contrary, it is not present in T cells, NK cells and plasma cells [3]. This protein plays an essential role in B cell lymphopoiesis, being important for development, maturation and differentiation of immature B-cell and also for proliferation and survival of B-cells themselves [4,5].

Btk is a fundamental component of different B cell receptors (BCR) and it acts as a modulator of several intracellular signals by a variety of cell surface molecules, including PI3K, MAPK and NF-κB pathways (Figure 1); in this way, it regulates activation, proliferation and differentiation to antibody-producing plasma cells [6,7]. Consequently, Btk inhibition leads to the interruption of many down-stream cell signaling pathways related to the development of B cell malignancies (e.g., different types of leukemias and lymphomas) [8] and autoimmune diseases (e.g., rheumatoid arthritis, RA, and multiple sclerosis, MS) [9]. 

Thus, Btk represents an important target in the drug development field, with 24 Btk inhibitors (BtkIs) currently under clinical evaluation as anti-tumor agents against chronic lymphocytic leukemia (CLL), small lymphocytic lymphoma (SLL), B-cell malignancies and mantle cell lymphoma (MCL) in different countries (i.e., USA, China and Poland) [10].

## 2. Btk Structure

As reported in Figure 1, Btk consists of five regions: the pleckstrin homology (PH) domain, the Tec homology (TH) domain, the Src homology (SH3) domain, the SH2 domain and the C-terminal region with kinase activity [11,12,13]. The PH domain mediates protein–phospholipid and protein–protein interactions. The TH domain is formed by two proline-rich regions (PRR) and is involved in autoregulation, whereas SH2 and SH3 domains bind phosphorylated tyrosine residues and PRR, respectively. In the SH3 domain, a fundamental autophosphorylated site (Y223 residue) is present. Finally, the C-terminal part contains the catalytic kinase domain embedding Y551 residue responsible for initial Btk activation [14]. In the catalytic domain, Cys481 residue represents the site of covalent binding of the most studied BtkIs.

In detail, Btk is activated by spleen tyrosine kinase (Syk), in turn activated by BCR [14]. Upon activation, Btk can phosphorylate Y753 and Y759 residues of PLCβ2, leading to stimulation and production of IP3, DAG and PKCβ. The levels of calcium increase and the MAPK/ERK path is triggered, affecting the transcriptional expression of genes involved in proliferation, survival and cytokine secretion. Simultaneously, Btk could activate Akt/Nf-kB signaling pathways [15,16]. Moreover, activated Btk is a mediator of pro-inflammatory signals, as inflammatory cytokines (TNFα, IL-1b), strictly associated with the inflammatory response [17].

## 3. Btk Inhibitors

Structural information about the kinase is fundamental for optimal inhibitor design. If the protein structure shows great plasticity, distinct ligands might induce different states of the kinase, as observed in multiple crystal structures in various conformations. 

The X-ray crystal structures of active and inactive Btk kinase bound to several inhibitors have been determined (PDB codes: 5P9J, 5P9H, 5P9M, 5P9L, 4OTF, 5P9F and 5P9G) [6]. The different ligands proved to induce different states of the kinase, providing interesting information and insights on Btk structure and function. 

According to their mechanism of action and binding mode, BtkIs can be classified into two types: (i) irreversible inhibitors characterized by a Michael acceptor moiety able to form a covalent bond with the conserved Cys481 residue in the ATP binding site; or (ii) reversible inhibitors that bind to a specific pocket in the SH3 domain through weak, reversible interactions (e.g., hydrogen bonds or hydrophobic interactions). In detail, they access the specific SH3 pocket of Btk, inducing an inactive conformation of the kinase. 

The majority of currently approved BtkIs are irreversible inhibitors (Table 1) [18]. However, the onset of resistant mutants (especially to ibrutinib, the first drug launched on the market) has reduced their use [19,20]. In particular, the isosteric replacement of Cys481 into a serine residue decreases the reactivity of Btk variant towards ibrutinib and the other covalent inhibitors with a reduction in compound potency. As an example, ibrutinib showed a sixfold reduction in potency against C481S mutant (IC_50_ = 4.6 nM) [21]. Furthermore, other site mutations involving both Cys481 (e.g., C481R, C481F, C481Y) [21,22] and the gatekeeper residue Thr474 (T474I, T474S and T474M) [23] have been recently evidenced. Although ibrutinib can still bind noncovalently to C481S mutant, its reversible mechanism of action does not guarantee therapeutic efficacy in patients with this mutation [24,25]. 

In this regard, non-covalent inhibitors that do not interact with Cys481 could inhibit C481R, T474I and T474M mutants and represent an interesting therapeutic option [21]. Moreover, to date, the use of reversible inhibitors seems to be more effective in treating autoimmune diseases such as RA, different types of MS, chronic graft versus host disease (cGVHD) and systemic lupus erythematosus (SLE) [26,27].

More recently, some proteolysis-targeting chimera (PROTAC) molecules have been reported as a new therapeutic approach to reduce Btk activity [28].

### 3.1. Approved Btk Inhibitors

The three BtkIs currently approved for clinical use are ibrutinib ((S)-1-(3-(4-amino-3-(4-phenoxyphenyl)-1H-pyrazolo [3,4-*d*]pyrimidin-1-yl)piperidin-1-yl)prop-2-en-1-one (Imbruvica^®^, Pharmacyclics LLC, Sunnyvale, CA, USA); acalabrutinib (4-((3S)-8-amino-3-((R)-1-(but-2-ynoyl)pyrrolidin-2-yl)-3,8a-dihydroimidazo[1,5-*a*]pyrazin-1-yl)-*N*-(pyridin-2-yl)benzamide (Calquence^®^, AstraZeneca Pharmaceuticals LP, Gaithersburg, DE, USA); and zanubrutinib (7-(1-acryloylpiperidin-4-yl)-2-(4-phenoxyphenyl)-4,5,6,7-tetrahydropyrazolo[1,5-*a*]pyrimidine-3-carboxamide (Brukinsa^®^, BeiGene USA, Inc., San Mateo, CA, USA) (Figure 2A) [29]. As highlighted in Figure 2, these compounds share some structural similarity, although they are characterized by different pyrazolo[3,4-*d*]pyrimidine, dihydroimidazo[1,5-*a*]pyrazine and tetrahydropyrazolo[1,5-*a*]pyrimidine scaffolds. In addition, ibrutinib and zanubrutinib present a common 4-phenoxyphenyl substituent at position 3 of the pyrazole nucleus and a piperidin-1-yl-prop-2-en-1-one chain, very similar to that of acalabrutinib. Furthermore, ibrutinib and acalabrutinib display a free amino group on the heteroaromatic core nucleus. On the basis of the crystallographic data available, ibrutinib and zanubrutinib share similar bioactive conformations within the wild-type Btk binding site (Figure 2B) [30,31]. Therefore, beside the covalent bond with Cys481, the two complexes are mainly stabilized by similar interactions which include the cation-π contact between the phenoxyphenyl ring and Lys430 side chain and the hydrogen bonds with Met477 and Glu475 backbones (Figure 2C,D). It is clearly demonstrated that covalent interaction is not required to generate a potent Btk inhibitor, but with the ability to trap the enzyme in a covalent dead end complex, covalent irreversible BtkIs have a great potency [21].

Ibrutinib (also named PCI-32765) is a first-in-class Btk inhibitor. After the failure of LFM-A13 in 1999 [32], ibrutinib was initially chosen for preclinical development of in vivo models of RA in 2007 [14]. In 2010, Honigberg and coworkers reported the efficacy of this compound in B-cell lymphoma [33] and subsequently, in 2013, it was approved by the FDA for the treatment of CLL, SLL, Waldenström’s macroglobulinemia (WM), marginal zone lymphoma (MZL) and relapsed/refractory MCL. In 2017, the compound received approval also for cGVHD patients after failure of one or more lines of systemic therapy (Table 1) [34]. Selectivity is an important factor influencing the long-term safety of a drug, but it is impossible to predict every off-target protein that a covalent inhibitor may bind. In addition to Btk, ibrutinib also inhibits other kinases that possess Cys481-like residues including Blk, Bmx, Egfr, ErbB2, ErbB4, Itk, Tec, Txk and Jak [33]. Interestingly, ibrutinib also potently inhibits kinases that lack a reactive cysteine, such as Csk, Fgr, Lck, Brk, Hck, Yes1, Frk, Ret, Flt3, Abl, Fyn, Lyn and Src [33]. A recent proteomic study showed that ibrutinib could also covalently react with non-kinase proteins in cells [35]. To overcome ibrutinib off-target side effects (i.e., skin and dermatological problems [36], bleeding, infection [37], headache and atrial fibrillation) and the emerging resistances [24,38,39], some selective second-generation BtkIs were developed. 

Acalabrutinib (ACP-196, Figure 2A) is a novel second-generation Btk inhibitor, designed by Acerta Pharma [40,41]. It was approved in 2017 and is currently indicated for patients with relapsed/refractory MCL as well as CLL/SLL (Table 1). Zanubrutinib (BGB-3111, Figure 2A) was developed by BeiGene in 2012 [42] as a potential candidate due to its high potency, selectivity, in vitro pharmacokinetics and pharmacodynamics in an OCI-LY10 DLBCL xenograft model [31]. It was approved in 2019 for patients with MCL who have received at least one prior therapy, becoming the first Chinese-origin drug that won a grand slam tournament in FDA history.

Although ibrutinib, acalabrutinib and zanubrutinib are irreversible inhibitors able to covalently bind cysteine 481 in the ATP binding pocket of Btk, their activity on the enzyme is quite different (IC_50_ values = 1.5, 5.1 and 0.5 nM for ibrutinib, acalabrutinib and zanubrutinib, respectively). Moreover, as previously reported, all three compounds are not selective for Btk, showing inhibition in the low nanomolar range of other type of intracellular (e.g., Tec, Itk Blk, Jak) and receptor (e.g., epidermal growth factor receptor, EGFR) tyrosine kinases. The lack of specificity is probably associated with rash and severe diarrhea [43]. In particular, acalabrutinib represents the most selective compound [44], being inactive on Itk, EGFR, ERBB2, Blk and Jak3. Zanubrutinib and acalabrutinib are less active than ibrutinib against Tec and Itk [45] and therefore show less platelet disfunction and bleeding problems [46] and antibody-dependent cell-mediated cytotoxicity [47]. In addition, thrombus formation was significantly inhibited in platelets treated with ibrutinib, whereas no impact on thrombus formation was identified upon treatment with acalabrutinib that therefore displays an improved safety profile with minimal adverse effects compared with ibrutinib [48]. Very recently, Fancher and coworkers reported an interesting study on drug–drug and drug–food interactions associated with ibrutinib, acalabrutinib and zanubrutinib, providing recommendations for their usage, particularly on dosage, in clinical use [29]. The therapeutic indications, dosage and the total number of clinical trials of ibrutinib, acalabrutinib and zanubrutinib are reported in Table 1. The majority of clinical trials are focused on diseases for which these compounds have been approved, but in the past few years, these molecules have been evaluated for their immunomodulation activity (as previously reported), with cGVHD and more recently COVID-19 being the principal objects of BtkI application. 

For cGVHD, a condition that might occur after an allogeneic transplant, the most studied compound is ibrutinib, with nine clinical trials carried out (one completed, five in recruitment, two active studies not recruiting, one enrolling by invitation, Table 1). In detail, ibrutinib has been evaluated in association with rituximab (NCT03689894 and NCT04235036) and corticosteroids (NCT02959944) and in a comparative study with ruxolitinib (NCT03112603). Additionally, two clinical trials (NCT04198922 and NCT04716075; Table 1) are currently recruiting participants for the evaluation of acalabrutinib for the treatment of cGVHD.

Despite the evidence of the beneficial effect in MS of anti-inflammatory agents [9,49], currently no clinical trials regarding the use of irreversible BtkIs in this pathology are reported.

Considering systemic hyper-inflammation, cytokine storm induced by COVID-19 infection and the active role in macrophage function at NF-kB pathways of Btk, in 2020, some clinical studies on hospitalized COVID-19 patients have been carried out on BtkIs [50]. To date, two clinical trials are reported for ibrutinib, three for acalabrutinib (two of them completed) and one for zanubrutinib (Table 1). Although all these data support the potential use of BtkIs in the COVID-19 treatment, the potential increased risk of secondary infections or impaired humoral immunity in patients should be considered; indeed, opportunistic infections (particularly pneumonia) are commonly reported in treated BtkI patients [37]. In detail, data from the CALAVI phase II trials (NCT04497948) for acalabrutinib in hospitalized COVID-19 patients did not meet the primary efficacy endpoints and, on the base of these results, this study has been prematurely terminated [51].

#### Drug Delivery of Ibrutinib

Nanotechnologies represent an effective approach to overcome the pharmacokinetic issues associated with small molecules (e.g., poor water solubility, limited oral bioavailability, large distribution volume) and the administration of a single nanoparticle (NP) containing several drugs proved to be more effective than the administration of several NPs each containing one compound [52,53].

The poor water solubility of ibrutinib limits its absorption and bioavailability, negatively effecting the drug’s therapeutic effect. Recently, to improve its efficacy in cancer therapy, different nanoformulations of ibrutinib have been developed, including gold and polymeric NPs or aqueous nanosuspension for oral administration. 

One of the first studies regarding ibrutinib delivery was focused on the innovative approach of targeting cancer cells with an increased cholesterol demand [54]. Gold nanoparticles functionalized with apolipoprotein A-I and a phospholipid bilayer (HDL NPs) were able to reduce cellular cholesterol uptake in B-cell lymphoma and synergize with inhibitors of downstream B-cell receptor signaling, including ibrutinib. The study demonstrated that the ABC lymphoma cell lines are more resistant to the reduction in cholesterol by HDL NPs, but the combination of these nanoparticles with ibrutinib (5 nM) significantly reduced total cellular cholesterol. The obtained data confirmed that cellular cholesterol depletion induces apoptosis in lymphoma cells and provided a rational approach to target cholesterol metabolism in other cancer types that are cholesterol-dependent. 

Sanchez-Coronilla and coworkers [55] presented a theoretical study with ibrutinib conjugated with cysteine/methyl-cysteine and gold surface. In particular, the interaction of the drug with a gold surface was studied to explore the possibility to use gold NPs as an ibrutinib delivery system. Based on the obtained results, the authors concluded that gold NPs could represent a valuable delivery system for ibrutinib that can interact with gold through the nitrogen atoms of the pyrimidine ring and the amino group. Interestingly, the ibrutinib acrylamide group would not be involved in the interaction with the gold surface and therefore can react with the Cys481 side chain, thus inhibiting the Btk enzyme.

Additionally, several polymeric NPs emerged to be very promising in enhancing ibrutinib action. Peng and coworkers [56] reported the preparation of cellulose derivative NPs obtained by conjugating 2,3-dialdehyde cellulose (DAC) with oleylamine and aminoethyl rhodamine (AERhB) via Schiff base bonds. AERhB was used as a model compound representative of amine-containing anticancer drugs, such as ibrutinib. Two kinds of NPs (namely, DAC-50% oleylamine/50% AERhB and DAC-75% oleylamine/25% AERhB), were used for the drug release studies under both physiological (pH = 7.4) and acid (pH = 5.0 and pH = 4.0) conditions to mimic the existing environment in cancerous tissues. After 48 h at 37 °C, the determined release percentages were 23.3%, 64.9% and 84.8% at pH 7.4, 5.0 and 4.0, respectively. These results indicated that the release of the model drug was predominantly driven by acid-induced degradation of the Schiff base linkages.

Qui and coworkers [57] described the preparation of self-assembled nanocomplexes constituted by sialic acid conjugated with stearic acid, ibrutinib and egg phosphatidylglycerol. The efficiency of this system in targeting macrophages and its efficacy in inhibiting tumor progression were investigated in vitro and in vivo. The results indicated that the nanocomplex exhibited high efficiency in targeting tumor-associated macrophages, inhibiting Btk activation and Th2 tumorigenic cytokine release, reducing angiogenesis and suppressing tumor growth. The authors claimed that the developed nanocomplexes could be a promising strategy for ibrutinib delivery with minimal systemic side effects.

Noteworthy, two papers reported ibrutinib formulations for oral administration. In particular, an aqueous nanosuspension containing ibrutinib and pluronic F-127 as stabilizing agent was optimized through a three-level, three-factor, Box–Behnken design [58]. The technological properties of the obtained nanosuspension (i.e., particle size between 278.6 and 453.2 nm, stability of freeze-drying formulation up to 6 months and controlled drug release) as well as the in vivo pharmacokinetic were properly defined. A second investigation considered the oral bioavailability and pharmacokinetics of poly(lactic-*co*-glycolic acid) NPs (PLGA-NPs) loaded with ibrutinib [59]. In details, PLGA is a polymer used for its biocompatibility, biodegradability and tunable physicochemical and formulation characteristics. After administration, it is transformed in lactic acid and glycolic acid, which are endogenous materials that do not cause immunogenic reactions. In this study, PLGA-NPs composed by 75% lactic acid and 25% glycolic acid (75:25) have been used to grant a slower degradation of the nanoparticles and a consequent sustained drug release. The authors observed 4.2-fold enhancements in the oral bioavailability of ibrutinib-loaded PLGA-NPs in comparison with an ibrutinib suspension, which could be attributed to better absorption and higher exposure of the nanoformulation. 

### 3.2. BtkIs under Clinical Investigation

In recent years, many molecules able to block Btk, both irreversibly and reversibly, have been patented and reported in the literature. In the past ten years [6,26,60], different chemical scaffolds (e.g., pyrimidines, 2,4-diaminopyrimidines, 1,3,5-triazines and condensed structures as pyrazolo-pyrimidines, pyrazolo-pyridines, pyrrolo-pyrimidines, pyrrolo-triazines, imidazo-pyrazines, imidazo-pyrimidines, imidazo-quinoxalines and purines) have been deeply investigated. In this review, we summarized the recently developed (ir)reversible BtkIs studied in clinical trials, covering recent advances in the field of medicinal chemistry. 

#### 3.2.1. Irreversible BtkIs

Spebrutinib [61], evobrutinib [62], olmutinib [63], tirabrutinib [64], elsubrutinib (ABBV-105) [65,66] and tolebrutinib (SAR 442168) [67] are irreversible BtkIs currently under clinical investigations (Figure 3, Table 2). These compounds share an α,β-unsaturated carbonyl moiety (essential for covalent bonds) and an aromatic ring (preferentially a pyrimidine nucleus), free or fused with other five-member cycles (Figure 3). Furthermore, with the sole exception of elsubrutinib, these derivatives are basic compounds that inhibit Btk at nanomolar concentrations (IC_50_ values of 0.5 nM, 37.9 nM, 1.0 nM and 2.2 nM for spebrutinib, evobrutinib, olmutinib and tirabrutinib, respectively) [26]. In March 2020, tirabrutinib was approved in Japan (at the dosage of 480 mg orally) for the treatment of recurrent or refractory primary central nervous system lymphoma and now is also under study for the treatment of WM, lymphoplasmacytic lymphoma and a number of autoimmune disorders (chronic lymphocytic leukemia, B cell lymphoma, Sjogren’s syndrome, pemphigus and RA) [68]. Olmutinib also inhibits in irreversible mode EGFR, a member of Tec family kinases (including Jak3, EGFR, Her2, Her4 and Blk) sharing with Btk a reactive cysteine residue (namely, Cys797) [69]. TG-1701 (TG Therapeutics), TAS5315 (Taiho Pharmaceutical) and M7583 (TL-895, EMD Serono) are irreversible BtkIs (molecular structures not disclosed) currently in clinical trials for B-cell malignancies (NCT03671590), RA (NCT03605251) and MCL (NCT02825836), respectively. In particular, TAS5315 is a pyrazolo[3,4-*d*]pyrimidine derivative currently in a phase II clinical trial for RA treatment [70]. In addition, DTRMWXHS-12 (also named DTRM-12, formula not disclosed) is a pyrazolo-pyrimidine derivative [71,72] acting as Btk irreversible inhibitor and currently under three phase I clinical trials for different types of leukemia (as CLL) and lymphoma (as MCL) (Table 2). The presence of reactive Michael acceptor groups in the above mentioned irreversible BtkIs led to undesired side effects such as allergic reactions, fever, lymphadenopathy, edema and albuminuria due to off-target inhibition [73]. 

#### 3.2.2. Reversible BtkIs

Noncovalent BtkIs offer several advantages over existing covalent inhibitors. Whereas covalent inhibitors loose potency against Cys481 mutants, some noncovalent inhibitors retain potent inhibition against C481S and C481R Btk variants, providing a potentially effective treatment option to ibrutinib-resistant or naïve patients [21]. Furthermore, reversible inhibitors provide a lower risk of toxicity compared to irreversible compounds and for these reasons some of them (i.e., fenebrutinib, vacabrutinib, BMS-986142, BIIB068, CT-1530, AC0058 and SHR1459) are under clinical investigations for long-term drug administration in the treatment of autoimmune diseases, especially RA (Table 2).

In the past ten years, a plethora of reversible BtkIs have been patented [6,26,61]; for example, Genentech corporation generated over 1000 noncovalent BtkIs covering a broad range of chemical substructures and physicochemical properties [21]. Substantial efforts have been made over the years to further develop reversible inhibitors, but unfortunately, none of the studied compounds have yielded significant breakthroughs [6,74,75]. The most recent and relevant reversible BtkIs are reported in Figure 4. 

In particular, fenebrutinib is a potent Btk inhibitor (IC_50_ = 0.5 nM) endowed with good selectivity, favorable pharmacokinetic profile and efficacy against Btk C4815 mutant, [76,77,78], whereas vacabrutinib potently inhibits Btk and Itk [79,80]. Fenebrutinib is currently under clinical evaluation in eight trials (i.e., NCT03693625, NCT03596632, NCT04586023, NCT04586010, NCT04544449, NCT01991184, NCT03137069, NCT02908100) focused in particular on autoimmune conditions, whereas the NCT03037645 trial assessed vacabrutinib activity in different hematological tumors (Table 2).

Pirimidinone derivative BMS-986142 (Figure 4) is currently under clinical studies for its activity in RA (NCT02880670, NCT02456844, NCT02762123, NCT02832180, NCT02638948) and Sjögren’s Syndrome (NCT02843659). Moreover, a clinical study (NCT02257151) on healthy adults has been completed. 

RN-486 displayed IC_50_ values of 4 nM, 43 nM and 64 nM for Btk, Slk and Tec, respectively [6,81,82]. Currently, this compound is in preclinical investigation for RA. Unfortunately, RN-486 as well as GDC-0834, developed from compound CGI-1746 and under study for arthritis treatment, evidenced poor stability and pharmacokinetic profiles [83,84]. 

BIIB068 demonstrated good kinome selectivity (IC_50_ = 1 nM for Btk) and good overall drug-like properties for oral dosing; it was well tolerated across preclinical species at pharmacologically relevant doses with good ADME properties and achieved >90% inhibition of Btk phosphorylation in humans [85,86]. BIIB068 seems to be effective in SLE disease (clinical study NCT02829541).

GNE-431 is a new and interesting molecule, able to inhibit C481R, T474I and T474M mutants [21,87], representing the first example of “Pan-Btk” inhibitors. In detail, GNE-431 showed IC_50_ = 3.2 nM against wild-type Btk and similar potency against C481S mutant (IC_50_ = 2.5 nM) [21]. To date, no clinical studies are reported for this compound. The widespread activity of GNE-431 against Btk mutants (namely, C481S, C481R, T474I and T474M) has been structurally rationalized by docking simulations [21]. The ligand would assume an extended conformation being oriented orthogonally in comparison with ibrutinib. The hexahydropyrazino[1,2-*a*]indol-1(2H)-one moiety of GNE-431 would be inserted in the H3 subpocket (a unique site in Btk) and its imidazopyridazine core would be able to interact with hydrophobic gatekeeper residues at position 474. Furthermore, GNE-431 would poorly interact with residue at position 481 and its binding mode would be marginally affected by the different steric requirements of residues at position 474.

Many other molecules are under clinical and preclinical studies (APQ531, SHR1459, CT-1530, AC0058), but their chemical structures are not disclosed.

#### 3.2.3. Emerging Reversible Covalent BtkIs

One approach for the discovery of next-generation BtkIs with high potency, enhanced selectivity profiles, reduced off-target effects as well as tunable residence times, is the design of compounds able to form reversible covalent bonds with Cys481 residue and temporarily inactivate the enzyme.

PRN1008 (rilzabrutinib; Figure 5, Table 2) is in early clinical studies for RA treatment [88,89]. PRN-1008 is a potent, selective and reversible covalent inhibitor of Btk (IC_50_ = 3.1 nM), inhibiting the kinase by forming a covalent bond with the Cys481 residue [90]. In vivo, PRN-1008 demonstrated enduring pharmacodynamic effects [6] and suppressed collagen-induced arthritis in rats in a dose-dependent manner. These data support the continued development of PRN-1008 as a therapeutic agent for RA. In addition, in 2017, Principia Biopharma announced that PRN-1008 had been designated as orphan drug in the USA for the treatment of pemphigus vulgaris, significantly reducing prednisone use and its related risks, and might become an important treatment option for this devastating disease [6].

Bradshaw and coworkers [91], using a structure-based design, developed a series of reversible, covalent Btk inhibitors related to PRN1008. These compounds embed a reversible cyanoacrylamide-based electrophile attached to a pyrazolo-pyrimidine scaffold via an amine containing heterocycle linker (piperidine or pyrrolidine). The cyanoacrylamide functionality is capped with various branched-alkyl groups with different steric and electronic properties. The nature of the capping groups affects the residence time of the reversible inhibitors. Compound **1** (Figure 5) was identified as a promising lead compound with sustained Btk occupancy. In the crystal structure of Btk-**1** complex (PDB code 4YHF) [91], the ligand is covalently bonded to Cys481 and the amino-pyrrolopyrimidine portion forms hydrogen bonds with Thr474, Glu475 and Met477 (Figure 5B). Interestingly, the *tert*-butyl group shield the proton attached to Cα, thus preventing the breakage of the thioether bond with Cys481. 

To enhance solubility and oral bioavailability of compound **1**, a series of methylpyrrolidine-containing compounds was further developed as reversible, covalent BtkIs. Compound **2** (Figure 5) exhibited an increased residence time, but dissociated rapidly and quantitatively upon Btk turnover, proteolysis, resynthesis and interaction with cellular binding partners. Additionally, derivative **2** emerged to be as effective as the irreversible covalent inhibitor ibrutinib in decreasing tumor cell invasiveness and blocking Btk activity [90,91].

This interesting new class of molecules combines the advantages of both reversible and irreversible binding mechanisms and shares a pyrrolopyrimidine scaffold that blocks the Btk active site through hydrogen bonds and hydrophobic interactions and a reactive, modified cyanoacrylamide electrophile, able to form a tunable covalent bond with the exposed Cys481 residue.

## 4. Conclusions

In recent years, Btk has emerged as a new target in medicinal chemistry and many BtkIs have been patented and reported in the literature. To date, only three irreversible BtkIs have been launched on the market to treat different types of leukemias and lymphomas, whereas reversible BtkIs are under clinical investigations for long-term drug administration in the treatment of autoimmune diseases, especially RA and MS. Furthermore, these compounds also find application in the treatment of cGVHD, autoimmune inflammation that might occur after an allogeneic transplant and for which currently there are no effective and resolving therapies. Most notable is the application of approved BtkIs in systemic hyper-inflammation and cytokine storm induced by COVID-19 infection, whose effectiveness is still under evaluation. Unfortunately, the onset of resistance to irreversible inhibitors (in particular ibrutinib) and of off-target side effects, in particular skin and dermatological problems, prompted the search for selective second-generation BtkIs with a lower risk of toxicity compared to irreversible compounds. Nanoformulations of ibrutinib (including gold, polymeric NPs and aqueous nanosuspension) showed improved efficacy in cancer therapy, reduced toxicity and ameliorated absorption and bioavailability, thus representing valid help in overcoming resistance and side effects onset of the irreversible BtkIs in clinical use. In addition, the design of compounds able to form reversible covalent bonds with Cys481 residue and temporarily inactivate the enzyme seems to be an innovative and interesting approach. In particular, tunable BTK irreversible inhibitors represent a promising class of compounds with reduced side effects. Additionally, tunable BTKIs would allow the use of this kinase inhibitors in non-oncologic therapeutic areas which require chronic treatment such as auto-immune disorders (i.e., RA, SLE and cGVHD). For all these reasons, despite the presence in the literature of many reviews and articles, the study on the Btk functions and therapeutic applications, as well as the discovery of new Btk inhibitors and innovative formulations of approved compounds, is still very fruitful and of great interest to both the academic community and the pharmaceutical industry.

## Figures and Tables

**Figure 1 ijms-22-07641-f001:**
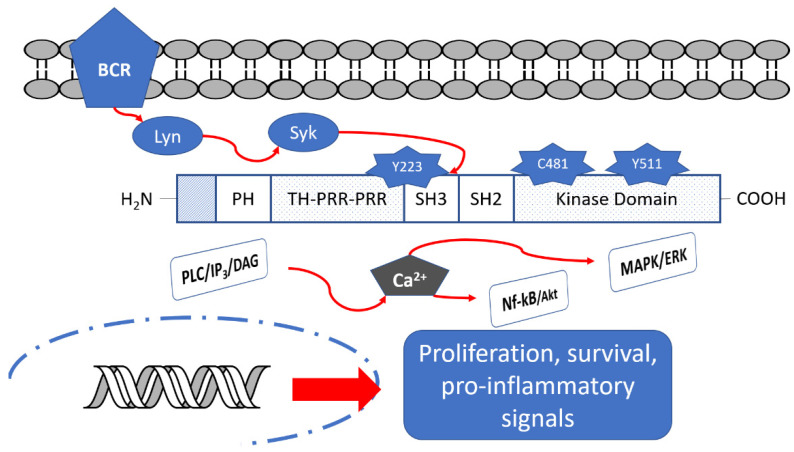
Btk structure and its down-stream signaling pathways.

**Figure 2 ijms-22-07641-f002:**
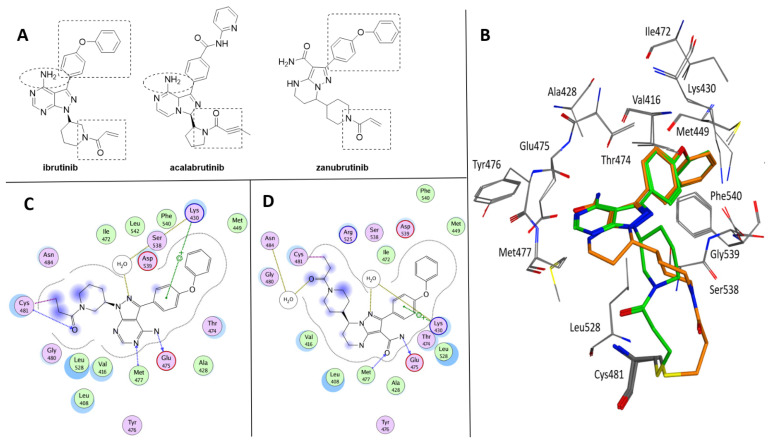
(**A**) Chemical structure of ibrutinib, acalabrutinib and zanubrutinib. The shared subportions are evidenced. (**B**) Superposition of Btk crystallized with ibrutinib (green, PDB code 5P9J) and zanubrutinib (orange, PDB code 6J6M). (**C**) Ligplot of the Btk–ibrutinib complex. (**D**) Ligplot of the Btk–zanubrutinib complex.

**Figure 3 ijms-22-07641-f003:**
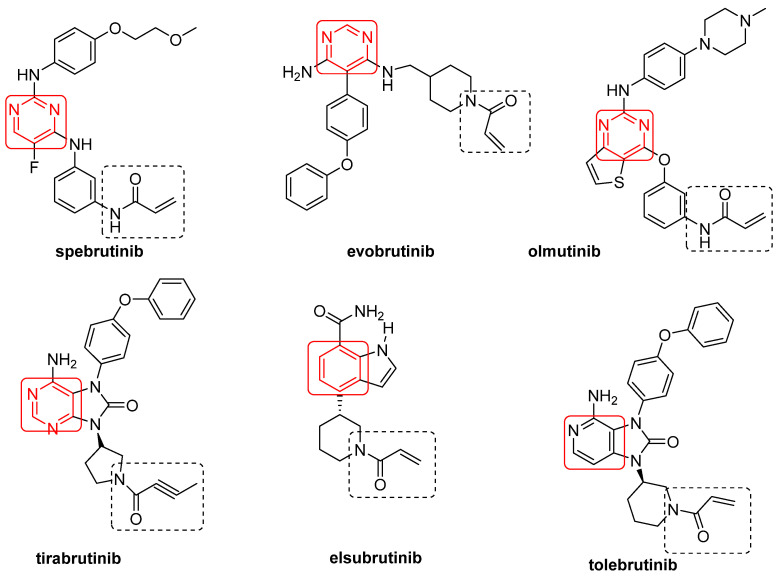
Irreversible BtkIs under clinical investigation.

**Figure 4 ijms-22-07641-f004:**
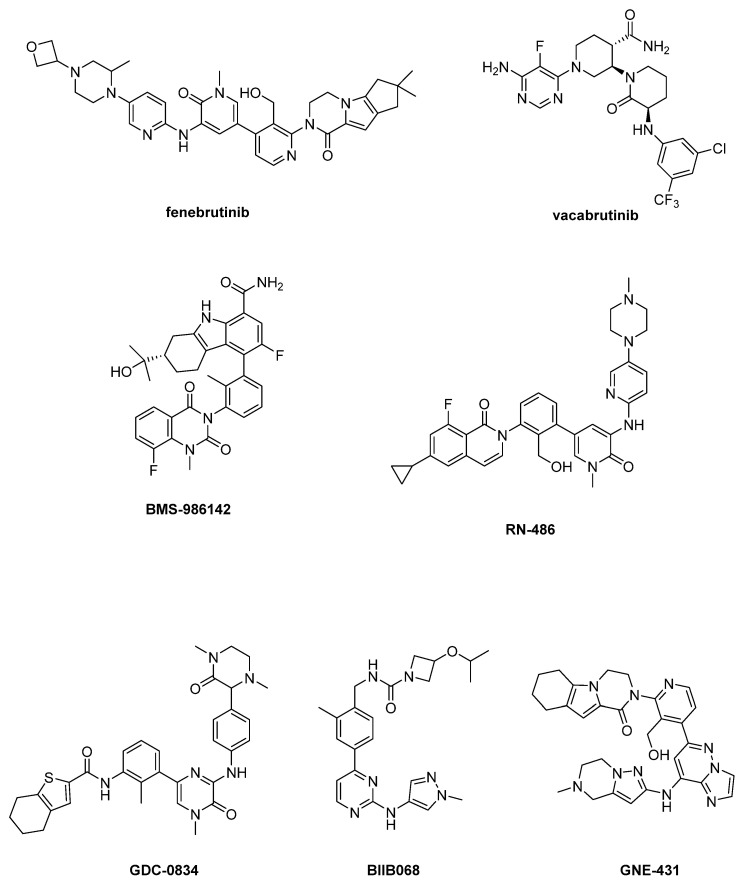
Reversible BtkIs under clinical investigation.

**Figure 5 ijms-22-07641-f005:**
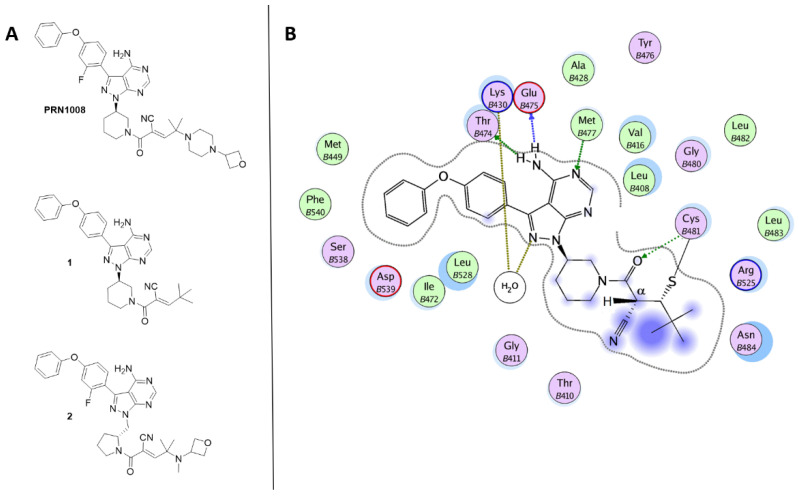
(**A**) Chemical structures of reversible covalent BtkIs PRN1008, 1 and 2. (**B**) Ligplot of the Btk-1 crystallographic complex.

**Table 1 ijms-22-07641-t001:** Dosage, clinical trials and treated diseases of approved BtkIs.

Drug	Dosage	Diseases	Number of Clinical Trials		
			Total	GVHD	COVID-19
Ibrutinib	420 mg/day (CLL/SLL, WM) 560 mg/day (MCL, MZL)	CLL/SLL WM MZL MCL cGVHD	358	9 (1 completed, 5 in recruitment, 2 actives not recruiting, 1 enrolling by invitation)	2 (NCT04375397 in recruitment; NCT04665115 not recruiting)
Acalabrutinib	100 mg b.i.d.	CLL/SLL MCL	99	2 (NCT04198922, NCT04716075, in recruitment)	3 (NCT04497948 terminated, NCT04380688 completed, NCT04346199 active not recruitment)
Zanubrutinib	160 mg b.i.d.	MCL	49	none	1 (NCT04382586 in recruitment)

**Table 2 ijms-22-07641-t002:** BtkIs under clinical investigation, number of clinical trials and treated diseases.

Compound	No. of Clinical Trials	Treated Diseases
Spebrutinib	8 (7 completed, 1 active, not in recruitment)	B Cell Non-Hodgkin’s Lymphoma, CLL, WM, RA
Evobrutinib	15 (4 in recruitment, the others completed or terminated)	MS, MSRR, RMS, RA, SLE
Olmutinib	8 (completed or terminated)	Different adenocarcinoma, NSCL, RA
Tirabrutinib	9	CLL, SLL, Non-Hodgkin’s Lymphoma, WM, RA, Sjogren’s Syndrome
Elsubrutinib	2, in recruitment	SLE
Tolebrutinib	7 (2 completed)	RMS, primary and secondary progressive MS
Fenebrutinib	8 (3 completed)	Urticaria, SLE, RMS, progressive MS, Lymphocytic Leukemia, lymphoma
Vacabrutinib	1 (terminated)	SLL, CLL
PRN1008 (Rilzabrutinib)	5 (1 completed)	RA, Immune Thrombocytopenia, Immune Thrombocytopenic Purpura, Pemphigus vulgaris
BMS-986142	7 (all terminated or completed)	RA, Sjögren’s Syndrome
CT-1530	1	Lymphoma, different autoimmune diseases
TG-1701	1 (recruiting)	Non-Hodgkin’s Lymphoma, CLL
AC0058	2 (1 completed)	SLE
SHR1459	7 (2 completed)	Lymphoma, different autoimmune diseases
RN-486	Preclinical investigation	
BIIB068	1 (completed)	SLE
DTRMWXHS-12	4 (2 completed, 1 in recruitment, 1 active not recruiting)	CLL and different leukemia and lymphoma

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
