# Peer review of "Btk Inhibitors: A Medicinal Chemistry and Drug Delivery Perspective"

_ijms, 2021, doi:10.3390/ijms22147641_

Round 1

Reviewer 1 Report

The work described in the present manuscript is consistent with the scope of the journal. Authors proposed Btk inhibitors: medicinal chemistry and drug delivery perspective. This work is methodically carried out and scientifically correct. There are few issues that the authors can address to improve their manuscript before acceptance for publication.

What is the novel in this work as compared to the previous literature?

The authors should add their opinions and suggestions to stimulate more studies in the field.

The authors should include one section on overview of reversible Btk inhibitors recently developed and studied in clinical trials.

The authors should include future perspective section.

Author Response

The work described in the present manuscript is consistent with the scope of the journal. Authors proposed Btk inhibitors: medicinal chemistry and drug delivery perspective. This work is methodically carried out and scientifically correct. There are few issues that the authors can address to improve their manuscript before acceptance for publication.

What is the novel in this work as compared to the previous literature?

BTKIs represent a well known class of anticancer drugs and their activity has been revised in a number of previously published papers [please see references 6, 18-20, 26-28]. As reported in abstract (lines 15-19), in the present review we present an update on the molecules currently under (pre)clinical investigation and highlight the relevance of nanotechnology to improve the activity of ibrutinib, the first approved BTK inhibitor.

The authors should add their opinions and suggestions to stimulate more studies in the field.

Lines 403-407 were added to the manuscript.

The authors should include one section on overview of reversible Btk inhibitors recently developed and studied in clinical trials.

Sections 3.2.2 and 3.2.3 of the submitted manuscript focus on the reversible BTK inhibitors currently under investigation in clinical trials. Table 2 summarized the clinical studies on these compounds. For sake of clarity, line 304 has been added and details on the clinical trial have been added in the text.

The authors should include future perspective section.

Lines 403-407 were added to the manuscript.

Reviewer 2 Report

The manuscript described (ir)reversible Bruton’s tyrosine kinase (Btk) inhibitors. Btk is involved in diseases such as cancers. Whereas irreversible inhibitors bind to Btk through a covalent bond, reversible inhibitors bind to it through hydrogen bonds or hydrophobic interactions. The majority of currently approved BtkIs are irreversible inhibitors. Reversible inhibitors are also wanted with respect to a variety of therapeutic approaches. The authors showed potent reversible Btkinhibitors.Thus, these findings will be useful for cancer therapy. Therefore, the manuscript is not too excellent to be published. In other words, the manuscript is so excellent that it should be published.

Comments
(1) It is thought that reversible inhibitors are easily discovered than irreversible inhibitors through random screening in vitro. Why are the majority of currently approved BtkIs irreversible inhibitors?
(2) Do free amino groups of ibrutinib and acalabrutinib show toxicities?
(3) Can compounds with SH group instead of double or triple bond form S-S bond with Cys481 to inhibit Btk? Is Btk-1 so?
(4) In line of 12, “others two irreversible Btk inhibitors” is preferable to be replaced with “two other irreversible Btk inhibitors”.
(5) In line of 15, “in the literature” is preferable to be replaced with “in the literatures”.
(6) In line of 24, “Tec Kinase family” is preferable to be replaced with “Tec kinase family”.
(7) In line of 26, “expressed by all cells” is preferable to be replaced with “expressed in all cells”.
(8) In line of 52, “responsible of” is probably preferable to be replaced with “responsible for”.
(9) In line of 81, “theirs use” is probably preferable to be replaced with “their use”.
(10) In line of 95, it is preferable to show what PROTAC stands for.
(11) In line of 128,“FDA” is probably preferable to be replaced with “the FDA”.

That is all.

Author Response

The manuscript described (ir)reversible Bruton’s tyrosine kinase (Btk) inhibitors. Btk is involved in diseases such as cancers. Whereas irreversible inhibitors bind to Btk through a covalent bond, reversible inhibitors bind to it through hydrogen bonds or hydrophobic interactions. The majority of currently approved BtkIs are irreversible inhibitors. Reversible inhibitors are also wanted with respect to a variety of therapeutic approaches. The authors showed potent reversible Btk inhibitors. Thus, these findings will be useful for cancer therapy. Therefore, the manuscript is not too excellent to be published. In other words, the manuscript is so excellent that it should be published.

Comments

(1) It is thought that reversible inhibitors are easily discovered than irreversible inhibitors through random screening in vitro. Why are the majority of currently approved BtkIs irreversible inhibitors?

The discovery of new reversible and irreversible BTKIs is a very long and difficult process. Up to date, only irreversible BTKIs have been approved on the basis of their strong activity against cancer and cGVHD. As stated in lines 119-121, the covalent interaction is not required to generate a potent Btk inhibitor but having the ability to trap the enzyme in a covalent dead end complex, covalent irreversible BtkIs have a great potency. A number of reversible compounds are under evaluation and hopefully, in a near future, a reversible BTK inhibitors will be approved and commercialized.

(2) Do free amino groups of ibrutinib and acalabrutinib show toxicities?

On the basis of the available information, the toxicities associated to ibrutinib and acalabrutinib (e.g. skin and dermatological problems, bleeding, infection, headache and atrial fibrillation) cannot be ascribable to the free amino group but to the irreversible binding with BTK. On the other hand, the free amino group is important for the interaction with BTK binding pocket (see figure 2C).

(3) Can compounds with SH group instead of double or triple bond form S-S bond with Cys481 to inhibit Btk? Is Btk-1 so?

In the BTK-1 complex (figure 5B), a covalent bond is formed between the cysteine-481 thiol group and the reactive a,b-unsaturated carbonyl moiety of the inhibitor. No disulfide S-S bond has been observed in the interaction between compound 1 and BTK.

(4) In line of 12, “others two irreversible Btk inhibitors” is preferable to be replaced with “two other irreversible Btk inhibitors”.

Manuscript has been amended as suggested.

(5) In line of 15, “in the literature” is preferable to be replaced with “in the literatures”.

Manuscript has been amended as suggested.

(6) In line of 24, “Tec Kinase family” is preferable to be replaced with “Tec kinase family”.

Manuscript has been amended as suggested.

(7) In line of 26, “expressed by all cells” is preferable to be replaced with “expressed in all cells”.

Manuscript has been amended as suggested.

(8) In line of 52, “responsible of” is probably preferable to be replaced with “responsible for”.

Manuscript has been amended as suggested.

(9) In line of 81, “theirs use” is probably preferable to be replaced with “their use”.

Manuscript has been amended as suggested.

(10) In line of 95, it is preferable to show what PROTAC stands for.

Manuscript has been amended as suggested.

(11) In line of 128, “FDA” is probably preferable to be replaced with “the FDA”.

Manuscript has been amended as suggested.

That is all.